# Patients Regret Their Choice of Therapy Significantly Less Frequently after Robot-Assisted Radical Prostatectomy as Opposed to Open Radical Prostatectomy: Patient-Reported Results of the Multicenter Cross-Sectional IMPROVE Study

**DOI:** 10.3390/cancers14215356

**Published:** 2022-10-30

**Authors:** Ingmar Wolff, Martin Burchardt, Christian Gilfrich, Julia Peter, Martin Baunacke, Christian Thomas, Johannes Huber, Rolf Gillitzer, Danijel Sikic, Christian Fiebig, Julie Steinestel, Paola Schifano, Niklas Löbig, Christian Bolenz, Florian A. Distler, Clemens Huettenbrink, Maximilian Janssen, David Schilling, Bara Barakat, Nina N. Harke, Christian Fuhrmann, Andreas Manseck, Robert Wagenhoffer, Ekkehard Geist, Lisa Blair, Jesco Pfitzenmaier, Bettina Reinhardt, Bernd Hoschke, Maximilian Burger, Johannes Bründl, Marco J. Schnabel, Matthias May

**Affiliations:** 1Department of Urology, University Medicine Greifswald, 17475 Greifswald, Germany; 2Department of Urology, St. Elisabeth Hospital Straubing, 94315 Straubing, Germany; 3Department of Urology, University Hospital Carl Gustav Carus, Technische Universität Dresden, 01307 Dresden, Germany; 4Department of Urology, Philipps-University Marburg, 35043 Marburg, Germany; 5Department of Urology, Klinikum Darmstadt, 64283 Darmstadt, Germany; 6Department of Urology and Pediatric Urology, University Hospital Erlangen, Friedrich-Alexander-University Erlangen-Nuremberg, 91054 Erlangen, Germany; 7Department of Urology, University Hospital Augsburg, 86156 Augsburg, Germany; 8Department of Urology, University Hospital Ulm, 89081 Ulm, Germany; 9Department of Urology, Paracelsus Medical University, 90419 Nuremberg, Germany; 10Department of Urology, Isarklinikum Hospital Munich, 80331 Munich, Germany; 11Department of Urology and Pediatric Urology, Hospital Viersen, 41747 Viersen, Germany; 12Department of Urology and Urologic Oncology, Hanover Medical School, 30625 Hanover, Germany; 13Department of Urology, Klinikum Ingolstadt, 85049 Ingolstadt, Germany; 14Department of Urology, Klinikum Neumarkt, 92318 Neumarkt Oberpfalz, Germany; 15Department of Urology, Evangelical Hospital Bethel, University Hospital Ostwestfalen-Lippe of the University Bielefeld, 33611 Bielefeld, Germany; 16Department of Urology and Pediatric Urology, Carl-Thiem-Klinikum Cottbus, 03048 Cottbus, Germany; 17Department of Urology, Caritas - St. Josef Medical Center, University of Regensburg, 93053 Regensburg, Germany

**Keywords:** prostate cancer, radical prostatectomy, clinical decision-making, decision regret, characterization of patients, survey

## Abstract

**Simple Summary:**

This multicenter study investigated the extent of patient’s decision regret (PatR) in patients with prostate cancer comparing different surgical modalities. Robot-assisted radical prostatectomy has replaced open radical prostatectomy as the surgical standard of care in many countries worldwide. However, a broad scientific basis evaluating the difference in patient-relevant outcomes between both approaches is still lacking. In this context, PatR is increasingly moving into the scientific focus. Our study shows a critical PatR in slightly more than one third of patients about 15 months after surgery. Patients who underwent robot-assisted surgery, and also patients without postoperative urinary stress incontinence, report significantly lower PatR. Likewise, this difference was also demonstrated for patients who decided together with their treating physician on the specific surgical procedure (consensual decision making). Our study helps to further establish PatR as an important endpoint in the setting of radical prostatectomy and identifies criteria which may be addressed to reduce PatR.

**Abstract:**

Patient’s regret (PatR) concerning the choice of therapy represents a crucial endpoint for treatment evaluation after radical prostatectomy (RP) for prostate cancer (PCA). This study aims to compare PatR following robot-assisted (RARP) and open surgical approach (ORP). A survey comprising perioperative-functional criteria was sent to 1000 patients in 20 German centers at a median of 15 months after RP. Surgery-related items were collected from participating centers. To calculate PatR differences between approaches, a multivariate regressive base model (MVBM) was established incorporating surgical approach and demographic, center-specific, and tumor-specific criteria not primarily affected by surgical approach. An extended model (MVEM) was further adjusted by variables potentially affected by surgical approach. PatR was based on five validated questions ranging 0–100 (cutoff >15 defined as critical PatR). The response rate was 75.0%. After exclusion of patients with laparoscopic RP or stage M1b/c, the study cohort comprised 277/365 ORP/RARP patients. ORP/RARP patients had a median PatR of 15/10 (*p* < 0.001) and 46.2%/28.1% had a PatR >15, respectively (*p* < 0.001). Based on the MVBM, RARP patients showed PatR >15 relative 46.8% less frequently (*p* < 0.001). Consensual decision making regarding surgical approach independently reduced PatR. With the MVEM, the independent impact of both surgical approach and of consensual decision making was confirmed. This study involving centers of different care levels showed significantly lower PatR following RARP.

## 1. Introduction

Radical prostatectomy (RP) represents a gold standard in the therapy of localized prostate cancer (PCA) [1,2,3]. A robot-assisted surgical approach (RARP) has become widespread, exceeding the number of traditional open surgical interventions (ORP). In the US, about 80% of RPs are nowadays performed using RARP [4,5]. In Germany, the proportion of RARP is continuously increasing and the tipping point was reached in 2018 when the number of RARPs exceeded the number of ORPs for the first time (Figure 1) [6].

Based on prospective studies comparing RARP and ORP, both approaches yielded similar results in terms of midterm oncological outcomes, urinary continence rates, and intraoperative complication rates [7,8,9,10]. Hu et al. found similar oncologic results for both approaches in their large retrospective study [4]. Lantz et al. reported better oncologic outcome parameters following RARP based on a longer follow-up, while a similar functional outcome in terms of urinary continence for both approaches was documented [11]. In the majority of available comparative studies, no essential differences concerning functional outcome with respect to urinary continence and potency rates were observed [7,8,11,12,13]. However, some studies found some advantages associated with RARP, especially in terms of erectile function [9,10,11,14]. Based on a matched-pair analysis, d’Altilia et al. demonstrated a higher proportion of patients with urinary incontinence following ORP compared to RARP [15]. In addition, soft criteria such as length of hospital stay and intraoperative blood loss have been shown to significantly improve with RARP as opposed to ORP [16,17,18].

In addition to these oncological and functional outcome parameters, in recent years patient-reported outcome measures have become of utmost importance in the treatment evaluation of patients undergoing cancer treatment [19,20,21,22,23,24,25]. Among these, patient’s regret (PatR) concerning the choice of therapy has been established as a relevant endpoint for the evaluation of treatment quality [21,26,27,28,29,30,31,32,33,34,35,36].

Wallis et al. found in their study analyzing 2072 patients five years after diagnosis of localized prostate cancer that PatR was higher in patients who underwent RP compared to patients under active surveillance, while no such difference could be demonstrated for local radiotherapy [35]. In this context, it remains unclear whether RARP has the potential to reduce PatR opposed to ORP as the small number of studies comparing RARP and ORP in this regard have resulted in conflicting data [37,38]. While Schroeck et al. demonstrated a significantly higher PatR in patients undergoing RARP opposed to ORP in their single center study, Baunacke et al. found no significant impact of the surgical approach on PatR in their German multicenter study [37,38]. However, both studies were conducted in the early era of RARP associated with a rather low proportion of patients undergoing RARP, so results may not easily be transferred to the present day.

Therefore, the aim of the “Importance of various supportive measures in the context of radical prostatectomy from the patient’s perspective study (IMPROVE study)” was to compare RARP and ORP with respect to patient-reported PatR within a large cohort in a contemporary multicenter setting.

## 2. Materials and Methods

From April to June 2021, a survey comprising personal, perioperative, and functional criteria was sent to a total of 1000 patients at a median of 15 months (IQR 11–20) after RP in 20 German urologic centers (50 patients per center). Patients who did not respond to the mailed surveys received one reminder.

Within the survey, urinary stress incontinence was evaluated with questions extracted from a validated questionnaire [39,40,41]. PatR was recorded using the Decision Regret Scale [42]. This scale is based on five questions with a five-item Likert-scale translated into a score of 0 (no regret) to 100 (highest regret) and has been validated for the evaluation of treatment options in prostate cancer [43]. Additionally, patients were asked to report the way clinical decision making regarding the surgical approach was carried out: passive decision (by physician alone), consensual decision making by patient and physician together, or active decision (by patient alone).

Based on a final questionnaire response rate of 75.0%, the data were merged with details related to surgical items provided by the participating centers. These included surgical approach, prostate-specific antigen (PSA) level prior to surgery, date of surgery, details concerning nerve preservation, TNM stage, surgical margin status, and complications during hospital stay recorded according to the Clavien–Dindo Classification (CDC) [44,45]. Finally, the center’s level of care (university vs. non-university) and the center’s mean caseload per year between 2018 and 2020 were recorded.

Patients who underwent laparoscopic RP (not robot-assisted) and those with distant metastatic disease were excluded from the study. Additionally, within this analysis, patients of one participating center had to be excluded due to missing data for different key criteria provided by the center.

A cutoff of >15 was defined as critical PatR in the majority of previous studies [31,37]. Therefore, to ensure comparability of study results with data previously published, the threshold defining critical PatR was again set at >15.

Continuous variables were reported as medians with interquartile ranges (IQRs). Pertinent characteristics of the patients undergoing surgery using these two surgical approaches (ORP and RARP) were compared using the Mann–Whitney test for not normally distributed variables, while categorical variables were compared using Fisher’s exact test and the Pearson Chi-square test.

Some patients in International Society of Urological Pathology (ISUP) group 1 (Gleason score of 3 + 3 = 6) based on final histopathology potentially may regret that they opted for surgery, especially in the case of complications or impaired functional outcome, as they might think retrospectively that active surveillance would have represented a better alternative. Therefore, the ISUP group was dichotomized into patients of ISUP group 1 and those with ISUP groups 2–5 prior to inclusion of this variable into multivariate models.

The pT stage was dichotomized into patients with cancer confined to the prostate (pT2) and those with PCA infiltrating beyond the prostate capsule (pT3-pT4). Complications according to CDC grades were dichotomized into patients not requiring additional interventions (CDC grades 0–2) and those with more severe complications (CDC grades 3–5). Urinary stress incontinence was dichotomized into patients with no incontinence, including patients who reported using a maximum of one safety pad per day, and those who stated that they used more than one pad per day.

To calculate the impact of the surgical approach (RARP vs. ORP) on a critical PatR exceeding 15, a multivariate logistic regression base model (MVBM) was established incorporating the surgical approach and additional independent variables not primarily affected by the surgical approach (age, personal relationship status, social security status, educational qualification, professional status, time interval between RP and survey, degree of patient’s involvement in clinical decision making regarding the surgical approach, center’s level of care, center’s mean RP caseload per year from 2018 until 2020, preoperative PSA level, ISUP group, pT-stage, and pN-stage).

Moreover, an extended model (MVEM) was additionally adjusted by criteria potentially affected by the surgical approach (surgical margin status, adjuvant local radiation treatment, postoperative urinary stress incontinence, nerve sparing, and postoperative complications according to CDC).

Using a “best-of-fit” approach, multivariate binary-logistic regression models were calculated with backward elimination of independent variables.

Statistical analyses were performed using IBM SPSS Statistics^®^ V28 (Armonk, NY, USA). Reported *p* values are two-sided with the statistical significance level set at *p* < 0.05.

The study was conducted in accordance with the Declaration of Helsinki and approved by the Leading Ethics Committee of Medizinische Hochschule Brandenburg (ethical approval E-01-20200805, date of approval: 17-August-2020). The IMPROVE study was registered in the German register of clinical studies (DRKS-ID: DRKS00023765).

## 3. Results

The survey was sent to patients after RP at a median of 15 months (IQR 11–20). After a maximum of one reminder, a response rate of 75.0% was achieved. For final analysis, the study cohort comprised 277 patients in the ORP group and 365 in the RARP group, resulting in an entire cohort of 642 patients who underwent RP between 2018 and 2020. Patient characteristics are shown in Table 1.

At a median interval of 15 months after RP, a median PatR of 15 (IQR 0–30) was found for ORP patients and of 10 (IQR 0–20) for RARP patients (*p* < 0.001). A total of 46.2% and 28.1% of patients in both groups were registered with a PatR exceeding the predefined cutoff of 15, respectively (*p* < 0.001).

Between the ORP and RARP groups, significant differences were observed regarding age, personal relationship (fixed vs. no fixed partnership), social security status (statutory vs. private health insurance), professional status (professionally active or professional activity scheduled vs. being retired), time interval between RP and survey, the degree of patient’s involvement in clinical decision making regarding the surgical approach, center’s level of care (university vs. non-university) and RP caseload per year, preoperative PSA level, pT sage, pN stage, surgical margin status, postoperative complication grades according to CDC, urinary stress incontinence, median decision regret, and proportion of patients reporting a decision regret exceeding the predefined cutoff of 15 (Table 1).

After backward elimination of all other initially included variables, the final MVBM to assess the independent impact of various parameters on critical PatR included surgical approach, age, professional status, degree of patient’s involvement in clinical decision making regarding the surgical approach, pT stage, and pN stage.

Based on this MVBM, patients who underwent RARP showed 46.8% less frequently a PatR exceeding the cutoff of 15 compared to ORP (OR: 0.532; *p* < 0.001) (Table 2). In addition, being retired (OR: 1.714; *p* = 0.030) and a consensually chosen surgical approach (physician and patient together as opposed to passive decision making, OR: 0.621; *p* = 0.027) were found to independently impact the endpoint (Table 2).

Finally, additional criteria potentially affected by the surgical approach were included in the MVEM. After backward elimination of all other initially included variables, final MVEM to assess the independent impact of various parameters on critical patient’s regret comprised surgical approach, age, professional status, degree of patient’s involvement in clinical decision making regarding the surgical approach, pN stage, and urinary stress incontinence.

Within MVEM, the independent impact of the surgical approach was confirmed (OR: 0.541; *p* = 0.001) (Table 3). Consensual decision making concerning the surgical approach was revealed to positively impact (OR: 0.608; *p* = 0.025), and being retired (OR: 1.692; *p* = 0.041) to negatively impact, the endpoint. Among the additionally included independent criteria, urinary stress incontinence was shown to be associated more than threefold more frequently with a critical PatR (OR: 3.292; *p* < 0.001). Moreover, based on MVEM, positive lymph node status (OR: 1.927; *p* = 0.028) and lower patient age (OR: 0.962; *p* = 0.032) were found to have a significant independent impact on critical PatR (Table 3).

## 4. Discussion

The IMPROVE study represents a large contemporary multicenter study evaluating the impact of surgical approach (ORP vs. RARP) on patient-reported PatR following radical prostatectomy for prostate cancer. Compared with other pertinent studies, an exceptionally high response rate of 75.0% was achieved enabling generalizability of our results [31,36,37,38].

Apparently, patients who underwent RARP were statistically significantly younger than those in the ORP group (66 vs. 69 years, *p* < 0.001). Additionally, patients in the RARP group were more likely to live in a fixed partnership and to have private health insurance, they were professionally active in a higher proportion, and they were involved in consensual clinical decision making regarding the surgical approach (patient and physician together) more frequently. Not surprisingly, RARP was performed more often in university centers and centers with a higher RP caseload per year, as these centers are equipped with a robotic system more frequently. Preoperative PSA levels differed significantly between ORP and RARP (8.6 ng/mL vs. 7.6 ng/mL, *p* = 0.030). However, it remains unclear whether this small difference represents a relevant clinical finding.

In contrast, the proportion of patients with locally advanced prostate cancer stages (pT3 + pT4) and lymph node involvement certainly represents a selection bias which occurred during clinical decision making regarding the surgical approach.

Interestingly, based on univariate analysis, in our cohort RARP was associated with a lower incidence of complications opposed to ORP (*p* < 0.001). Of note, in neither of the two groups were there any complications of grade 4–5 according to CDC reported within this study.

Notable urinary stress incontinence (>1 safety pad per day) was shown to be reported significantly less frequently in the RARP compared to the ORP group based on univariate analysis (14.6% vs. 24.9%; *p* = 0.001), which supports the results of d’Altilia et al. [15].

A median PatR of 15 (IQR 0–30) was found for ORP patients and of 10 (IQR 0–20) for RARP patients (*p* < 0.001), which confirms similar findings in other studies [31,36,37]. Lindsay et al. revealed a mean decision regret of 11.3 in their study including patients who underwent RARP, and Baunacke et al. found a mean decision regret of 14 in their entire cohort of 936 patients who underwent either ORP or RARP [31,37]. A total of 46.2% and 28.1% of patients were registered with a PatR exceeding the predefined critical cutoff of 15 in the ORP and the RARP groups, respectively (*p* < 0.001). This corresponds to data published by Lindsay et al. who reported 30% of patients with high decision regret after RARP [31].

In an effort to take all the differences between the two groups into account, two different multivariate models were established to assess the independent impact of the surgical approach on PatR.

Based on MVBM, patients who underwent RARP showed 46.8% less frequently a critical PatR (OR: 0.532; *p* < 0.001). Being retired (OR: 1.714; *p* = 0.030) and a consensually chosen surgical approach (OR: 0.621; *p* = 0.027) were also found to independently impact the endpoint parameter PatR.

Within MVEM, the independent impact of the surgical approach was confirmed (OR: 0.541; *p* = 0.001). Consensual decision making concerning the surgical approach was associated less frequently with a critical PatR (OR: 0.608; *p* = 0.025), while being retired (OR: 1.692; *p* = 0.041) had a negative impact on PatR. Among the additionally included independent criteria, urinary stress incontinence was shown to be associated more than threefold more frequently with a critical PatR (OR: 3.292; *p* < 0.001). Moreover, based on MVEM, positive lymph node status (OR: 1.927; *p* = 0.028) and lower patient age (OR: 0.962; *p* = 0.032) were found to have a significant independent impact on PatR.

The independent impact of the surgical approach (RARP vs. ORP) on a low PatR contradicts the findings of earlier studies. Schroeck et al. demonstrated a significantly higher PatR in patients undergoing RARP opposed to ORP in their single center study of 400 patients [38]. Moreover, lower urinary domain scores, lower hormonal domain scores, and the number of years since surgery were predictive for a high PatR. African American race and lower bowel domain scores were also identified as independent predictors of regret [38]. Baunacke et al. evaluated PatR based on a final study cohort comprising 404 patients who underwent RARP and 532 patients after ORP. After a follow-up of 6 years, they found no independent impact of patient’s age and surgical approach on PatR, while urinary continence, erectile function, absence of disease recurrence, shorter follow-up, and active decision making regarding treatment decisions were independently associated with a low PatR [37].

However, both studies were conducted in the early era of RARP and were potentially associated with excessively high expectations regarding the results of RARP among patients. In contrast, the early years of this new technique may have been affected by shortcomings connected with the establishment of robot-assisted surgical interventions, e.g., the learning curves of surgeons and staff. A change in surgical techniques over time and the higher proportion of patients undergoing RARP nowadays may also have contributed to the lower PatR of patients in the RARP group (opposed to the ORP group) in our study.

In contrast to the two pertinent studies mentioned before, no significant impact of time interval between RP and assessment of patient-reported outcome on PatR was found in the present study, and this variable has been removed from both multivariate models within the process of backward elimination of independent variables [37,38].

The independent impact of consensual clinical decision making regarding the surgical approach (patient and physician together), as opposed to a passive decision by the physician alone, was demonstrated in both multivariate models of this study and has also been found in other pertinent studies [29,37,46]. This finding underlines the importance of encouraging patients to participate in clinical decision making regarding the surgical approach. In addition, providing realistic outcome predictions prior to surgery seems of utmost importance in this context. Future research may clarify whether an intensified and standardized counselling of patients aiming at the active involvement of patients in treatment decisions may further reduce PatR following RARP.

Based on both multivariate models in our study, the professional status of being retired, as opposed to professional activity or professional activity scheduled again, was an independent predictor of a critical PatR. To our knowledge, this finding has not been reported in the literature so far. To date, one can only speculate about reasons. Perhaps retired patients have more time to reflect on their medical condition including potential harms following their treatment decisions.

Initially, we suggested that patients of ISUP group 1 (as opposed to groups 2–5) report a critical regret more often, as multiple other treatment options including active surveillance would have been available in their specific case. However, no such impact on the endpoint was found. Of interest, based on MVEM, patients with lymph node metastases were independently associated with a critical regret. This might reflect that physicians failed to familiarize patients presenting with aggressive tumors with the concept of multimodal therapy including RP as a first step.

This study is the first to demonstrate a lower PatR following RARP (opposed to ORP) for prostate cancer. Although the impact of the surgical approach on PatR was evaluated with thorough adjustments for potential confounders, our study has some limitations which have to be considered.

The different time intervals between RP and assessment of patient-reported outcome in the two groups could possibly have hampered interpretation of our results, as Hurwitz et al. reported increasing PatR over time in patients who underwent RP [29]. Therefore, the time interval between RP and survey was initially included into both multivariate models to account for this possible shortcoming. Interestingly, as mentioned above, based on both multivariate models within this study, the time interval between RP and survey was not significantly associated with PatR. Nevertheless, it represents a drawback that PatR has not been assessed at various predefined points in time, but only once at a median of 15 months after RP (IQR: 11–20).

Additionally, impairment of sexual function following RP was not assessed in this study. Therefore, results might be biased as different studies have shown an independent impact of sexual function on PatR following surgical treatment for prostate cancer [37,46]. However, no assessment of erectile function prior to surgery was available in a relevant number of participating centers. Unfortunately, a retrospective assessment of preoperative erectile function would not have provided reliable data. Therefore, no such evaluation was included in our questionnaire mailed to patients at a median of 15 months after RP because only the impairment of sexual function opposed to the status before surgery would have represented a scientifically relevant criterion. Of note, the extent of intraoperative nerve sparing (no nerve sparing vs. unilateral or bilateral nerve sparing) was included in the analysis.

Finally, both groups in this study (ORP and RARP) differed significantly in terms of several demographic characteristics, e.g., age, social security status, professional status, pT stage, and center specific variables such as level of care and mean annual RP caseload. However, in an effort to assess the independent impact of the surgical approach on PatR, two multivariate models were established adjusting for these possible confounders.

In summary, being based on a robust statistical analysis, this study represents a milestone in the assessment of patient-reported outcomes following RARP.

## 5. Conclusions

In this multicenter study involving German urologic centers with different levels of care, PatR was significantly lower after RARP compared to ORP. Additionally, an independent impact of consensual clinical decision making regarding the surgical approach (patient and physician together) on PatR was demonstrated. Finally, a relevant urinary stress incontinence, which was reported in 19.1% of the entire study cohort at a median of 15 months after RP, significantly increased PatR.

## Figures and Tables

**Figure 1 cancers-14-05356-f001:**
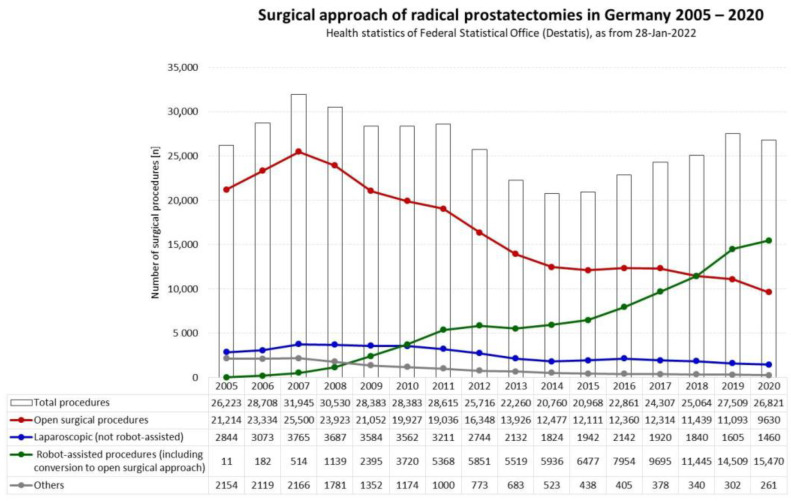
Trends concerning surgical approaches for radical prostatectomies in Germany 2005–2020 [6].

**Table 1 cancers-14-05356-t001:** Descriptive characteristics of 642 patients who underwent RP for PCA.

Variable	Entire Cohort (n = 642)	ORP Group (n = 277)	RARP Group (n = 365)	*p* Value
Age (n = 642): median (IQR) in years	68 (63–72)	69 (65–73)	66 (62–71)	<0.001
Personal relationship status (n = 639): fixed partnership no fixed partnership	586 (91.7%) 53 (8.3%)	245 (89.1%) 30 (10.9%)	341 (93.7%) 23 (6.3%)	0.043
Social security status (n = 642): statutory health insurance private health insurance	452 (70.4%) 190 (29.6%)	216 (78.0%) 61 (22.0%)	236 (64.7%) 129 (35.3%)	<0.001
Educational qualification (n = 639): university or technical college degree no such qualification	219 (34.3%) 420 (65.7%)	87 (31.6%) 188 (68.4%)	132 (36.3%) 232 (63.7%)	0.239
Professional status (n = 637): professionally active or professional activity scheduled again retired	176 (27.6%) 461 (72.4%)	53 (19.3%) 221 (80.7%)	123 (33.9%) 240 (66.1%)	<0.001
Time interval between RP and survey in month (IQR) (n = 642)	15 (11–20)	19 (15–22)	12 (10–15)	<0.001
Clinical decision-making regarding surgical approach (n = 636): Decision by physician alone (passive decision) Consensual (patient and physician together) Decision by patient alone (active decision)	150 (23.6%) 304 (47.8%) 182 (28.6%)	87 (32.0%) 133 (48.9%) 52 (19.1%)	63 (17.3%) 171 (47.0%) 130 (35.7%)	<0.001
Center’s level of care: non-university center university (n = 642)	369 (57.5%) 273 (42.5%)	208 (75.1%) 69 (24.9%)	161 (44.1%) 204 (55.9%)	<0.001
Center’s mean RP-caseload per year 2018-2020 (IQR) (n = 642)	95 (56–134)	52 (21–92)	125 (95–150)	<0.001
Preoperative PSA level in ng/mL (IQR) (n = 642)	8.0 (5.6–12.1)	8.6 (5.6–13.5)	7.6 (5.6–11.8)	0.030
ISUP group 1 (Gleason score 3 + 3 = 6) ISUP group 2 (Gleason score 3 + 4 = 7a) ISUP group 3 (Gleason score 4 + 3 = 7b) ISUP group 4 (Gleason score 4 + 4 = 8, 3 + 5 = 8, and 5 + 3 = 8) ISUP group 5 (Gleason score 4 + 5 = 9, 5 + 4 = 9, and 5 + 5 = 10) (n = 642)	53 (8.2%) 305 (47.5%) 163 (25.4%) 41 (6.4%) 80 (12.5%)	18 (6.5%) 133 (48.0%) 69 (24.9%) 14 (5.1%) 43 (15.5%)	35 (9.6%) 172 (47.1%) 94 (25.8%) 27 (7.4%) 37 (10.1%)	0.141
pT stage (n = 642): pT2a pT2b pT2c pT3a pT3b pT4	53 (8.3%) 7 (1.1%) 344 (53.6%) 130 (20.2%) 105 (16.3%) 3 (0.5%)	25 (9.0%) 5 (1.8%) 123 (44.4%) 62 (22.4%) 60 (21.7%) 2 (0.7%)	28 (7.7%) 2 (0.5%) 221 (60.6%) 68 (18.6%) 45 (12.3%) 1 (0.3%)	0.001
pN stage (n = 641): pN0+pNx pN1	580 (90.5%) 61 (9.5%)	234 (84.5%) 43 (15.5%)	346 (95.1%) 18 (4.9%)	<0.001
Surgical margin status (n = 642): R0 R1	477 (74.3%) 165 (25.7%)	194 (70.0%) 83 (30.0%)	283 (77.5%) 82 (22.5%)	0.036
No adjuvant local radiation adjuvant local radiation (n = 638)	521 (81.7%) 117 (18.3%)	218 (79.0%) 58 (21.0%)	303 (83.7%) 59 (16.3%)	0.148
Nerve sparing (n = 642): no nerve sparing unilateral nerve sparing bilateral nerve sparing	234 (36.5%) 106 (16.5%) 302 (47.0%)	107 (38.6%) 42 (15.2%) 128 (46.2%)	127 (34.8%) 64 (17.5%) 174 (47.7%)	0.537
Postoperative complications according to CDC grades (n = 642): 0 1 1d 2 3a 3b 4a, 4b, and 5	434 (67.6%) 63 (9.8%) 59 (9.2%) 49 (7.6%) 20 (3.1%) 17 (2.7%) 0	154 (55.6%) 42 (15.2%) 30 (10.8%) 36 (13.0%) 9 (3.2%) 6 (2.2%) 0	280 (76.7%) 21 (5.8%) 29 (7.9%) 13 (3.6%) 11 (3.0%) 11 (3.0%) 0	<0.001
Urinary stress incontinence (n = 639): safety pad/day >1 pad/day	517 (80.9%) 122 (19.1%)	208 (75.1%) 69 (24.9%)	309 (85.4%) 53 (14.6%)	0.001
Median decision regret of patients (IQR) (n = 632)	10 (0–20)	15 (0–30)	10 (0–20)	<0.001
Patient’s decision regret 0–15 Patient’s decision regret >15 (n = 632)	405 (64.1%) 227 (35.9%)	147 (53.8%) 126 (46.2%)	258 (71.9%) 101 (28.1%)	<0.001

Legend: CDC, Clavien–Dindo classification; IQR, interquartile range; ISUP, International Society of Urological Pathology; ORP, open surgical radical prostatectomy; PSA, prostate-specific antigen; RARP, robot-assisted radical prostatectomy; RP, radical prostatectomy.

**Table 2 cancers-14-05356-t002:** Multivariate logistic regression base model based on 642 patients who underwent RP assessing the independent impact of surgical approach, and various clinical and histopathological features not primarily affected by the surgical approach, on a critical patient’s decision regret (patient’s decision regret >15).

Variable	OR (95% CI)	*p* Value
**Surgical approach: RARP (reference: ORP)**	0.532 (0.370–0.765)	<0.001
**Age in years (continuous)**	0.970 (0.938–1.003)	0.072
Personal relationship status: no fixed partnership (reference: fixed partnership)	-	-
Social security status: private health insurance (reference: statutory health insurance)	-	-
Educational qualification: university or technical college degree (reference: no such qualification)	-	-
**Professional status: being retired** **(reference: professionally active or professional activity scheduled again)**	1.714 (1.052–2.792)	0.030
Time interval between RP and survey in month (continuous)	-	-
**Clinical decision-making regarding surgical approach** - **Consensual decision-making** **(patient and physician together)** - **Decision by patient alone** **(reference: decision by physician alone)**	0.621 (0.407–0.948) 0.735 (0.456–1.185)	0.027 0.206
Center’s level of care: University center (reference: non-university center)	-	-
Center’s mean RP-caseload per year 2018–2020 (continuous)	-	-
Preoperative PSA level in ng/ml (continuous)	-	-
ISUP group 2–5 (reference: ISUP group 1)	-	-
**pT stage: pT3-4 (reference: pT1-2)**	1.391 (0.954–2.028)	0.860
**pN stage: pN1 (reference: pN0 or pNx)**	1.660 (0.908–3.036)	0.100

Legend: ISUP, International Society of Urological Pathology; OR, odds ratio; ORP, open surgical radical prostatectomy; PSA, prostate-specific antigen; RARP, robot-assisted radical prostatectomy; RP, radical prostatectomy.

**Table 3 cancers-14-05356-t003:** Multivariate logistic regression extended model based on 642 patients who underwent RP assessing the independent impact of surgical approach, various clinical and histopathological features, and additional criteria potentially affected by the surgical approach, on a critical patient’s decision regret (patient’s decision regret >15).

Variable	OR (95% CI)	*p* Value
**Surgical approach: RARP (reference: ORP)**	0.541 (0.373–0.786)	0.001
**Age in years (continuous)**	0.962 (0.929–0.997)	0.032
Personal relationship status: no fixed partnership (reference: fixed partnership)	-	-
Social security status: private health insurance (reference: statutory health insurance)	-	-
Educational qualification: university or technical college degree (reference: no such qualification)	-	-
**Professional status: being retired** **(reference: professionally active or professional activity scheduled again)**	1.692 (1.021–2.805)	0.041
Time interval between RP and survey in month (continuous)	-	-
**Clinical decision-making regarding surgical approach** - **Consensual decision-making** **(patient and physician together)** - **Decision by patient alone** **(reference: decision by physician alone)**	0.608 (0.393–0.940) 0.765 (0.469–1.249)	0.025 0.284
Center’s level of care: University center (reference: non-university center)	-	-
Center’s mean RP-caseload per year 2018–2020 (continuous)	-	-
Preoperative PSA level in ng/ml (continuous)	-	-
ISUP group 2–5 (reference: ISUP group 1)	-	-
pT stage: pT3-4 (reference: pT1-2)	-	-
**pN stage: pN1 (reference: pN0 or pNx)**	1.927 (1.072–3.464)	0.028
Surgical margin status: R1 (reference: R0)	-	-
Adjuvant local radiation treatment (reference: no adjuvant local radiation treatment)	-	-
Nerve sparing: uni- or bilateral nerve sparing (reference: no nerve sparing)	-	-
Postoperative complications 3–5 according to CDC (reference: 0–2)	-	-
**Urinary stress incontinence: >1 pad per day** **(reference: 0–1 safety pad per day)**	3.292 (2.125–5.100)	<0.001

Legend: CDC, Clavien–Dindo classification; ISUP, International Society of Urological Pathology; OR, odds ratio; ORP, open surgical radical prostatectomy; PSA, prostate-specific antigen; RARP, robot-assisted radical prostatectomy; RP, radical prostatectomy.

## Data Availability

Not applicable.

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
