# Peer review of "Patients Regret Their Choice of Therapy Significantly Less Frequently after Robot-Assisted Radical Prostatectomy as Opposed to Open Radical Prostatectomy: Patient-Reported Results of the Multicenter Cross-Sectional IMPROVE Study"

_cancers, 2022, doi:10.3390/cancers14215356_

Round 1

Reviewer 1 Report

The authors documented the patient regret their choice of therapy significantly less frequently after RARP as opposed ORP. Reasonable results derived from a sufficient number of clinical samples are presented. The results are acceptable and well discussed.

Author Response

The authors documented the patient regret their choice of therapy significantly less frequently after RARP as opposed ORP. Reasonable results derived from a sufficient number of clinical samples are presented. The results are acceptable and well discussed.

Reply: We would like to thank the reviewer for the favorable assessment of our manuscript.

Reviewer 2 Report

This is a an interesting and important study. I congratulate the authors for successfully evaluating a patient related outcome, which has been frequently overlooked in clinical trials. While there are limitations, as mentioned by the authors due to the retrospective design of the study (Especially paucity of data in terms of sexual function), I still think, the findings of this study are practically valuable for both the patients and surgeon. The multivariate analysis reported has taken into account the appropriate factors and this adds to the validity of the findings. I do not have any changes to suggest interns of the design of the study or the analysis. 

Author Response

This is an interesting and important study. I congratulate the authors for successfully evaluating a patient related outcome, which has been frequently overlooked in clinical trials. While there are limitations, as mentioned by the authors due to the retrospective design of the study (Especially paucity of data in terms of sexual function), I still think, the findings of this study are practically valuable for both the patients and surgeon. The multivariate analysis reported has taken into account the appropriate factors and this adds to the validity of the findings. I do not have any changes to suggest interns of the design of the study or the analysis. 

Reply: We would like to thank the reviewer for the benevolent comments on our study.

Reviewer 3 Report

This is an elegantly written paper concerning patient reported outcomes (namely patient reported decision regret - PatR) after open versus robotic radical prostatectomy due to prostate cancer. I enjoyed the well thought-out study design and a thorough statistical analysis. Congratulations. This multi center analysis included 642 patients who responded to questionnaire invitation at a median of 15 months post RP. The authors found that patients after RARP are less likely to express significant PatR, compared to patients after ORP (28.1 vs 46.2% respectively).

The concerns regarding this study are somewhat addressed in discussion. The populations of patients undergoing RARP and ORP are quite disparate which makes selection bias likely. Patients who underwent  ORP were generally older, were more likely to rely on physician in decision-making process, and significant proportion suffered from higher-stage disease, which apart from surgical technique, may have contributed to inferior functional outcome. On the other hand, patients treated with RARP were more likely to be privately-insured, professionally active and more proactive in treatment choices, which resulted in treatment delivery in high-volume centers. Therefore this study compares not only ORP versus RARP with regard to PatR, but rather two distinct treatment philosophies: the older-style paternalistic still seen in small-volume departments doing ORPs in the second decade of 21st century, versus more contemporary, inclusive, multidisciplinary and shared-decision making-based typical to larger academic institutions, usually equipped with robotic systems. My feeling is this is worth commenting on.

Author Response

This is an elegantly written paper concerning patient reported outcomes (namely patient reported decision regret - PatR) after open versus robotic radical prostatectomy due to prostate cancer. I enjoyed the well thought-out study design and a thorough statistical analysis. Congratulations. This multi center analysis included 642 patients who responded to questionnaire invitation at a median of 15 months post RP. The authors found that patients after RARP are less likely to express significant PatR, compared to patients after ORP (28.1 vs 46.2% respectively).

Reply: We would like to thank the reviewer for the generous remark concerning our study design and for the suggestions listed below.  

The concerns regarding this study are somewhat addressed in discussion. The populations of patients undergoing RARP and ORP are quite disparate which makes selection bias likely. Patients who underwent  ORP were generally older, were more likely to rely on physician in decision-making process, and significant proportion suffered from higher-stage disease, which apart from surgical technique, may have contributed to inferior functional outcome. On the other hand, patients treated with RARP were more likely to be privately-insured, professionally active and more proactive in treatment choices, which resulted in treatment delivery in high-volume centers. Therefore this study compares not only ORP versus RARP with regard to PatR, but rather two distinct treatment philosophies: the older-style paternalistic still seen in small-volume departments doing ORPs in the second decade of 21st century, versus more contemporary, inclusive, multidisciplinary and shared-decision making-based typical to larger academic institutions, usually equipped with robotic systems. My feeling is this is worth commenting on.

Reply: Thanks to the reviewer for this helpful advice. A paragraph has been added to the discussion section to describe this limitation of our study and to explain the methods that were used to consider possible confounders within statistical analysis (lines 448-453).

Reviewer 4 Report

The manuscript presents very thorough analysis of factors determining patient regret following radical prostatectomy. The authors identify that surgical approach is an independent factor in lowering patient regret rate, namely that patients undergoing robot-assisted surgery have a higher long-term satisfaction regarding choice of therapy. The text is well written and easy to follow, with a detailed analysis of possible cofounding factors. The limitations of the study are represented by the lack of evaluation in sexual dysfunction following radical prostatectomy, as the authors also state.

Some minor comments regarding aspects that could be improved: 1) provide 1-2 sentences to better explain how the cuttoff value of 15 was chosen for patient regret, although it is detailed in the refferenced articles, ensuring some additional information regarding this aspext would improve the quality of the methods section; 2) the abbreviation for Clavien-Dindo Classification is not mentioned at the first use on line 151, although it is used in the following paragraphs as CDC; 3) the explanation for ISUP is provided on line 191, please move this on line 169 where it is first used. 

Author Response

The manuscript presents very thorough analysis of factors determining patient regret following radical prostatectomy. The authors identify that surgical approach is an independent factor in lowering patient regret rate, namely that patients undergoing robot-assisted surgery have a higher long-term satisfaction regarding choice of therapy. The text is well written and easy to follow, with a detailed analysis of possible cofounding factors. The limitations of the study are represented by the lack of evaluation in sexual dysfunction following radical prostatectomy, as the authors also state.

Reply: We would like to thank the reviewer for the complaisant assessment of our study highlighting the detailed analysis of possible confounding factors.

Some minor comments regarding aspects that could be improved: 1) provide 1-2 sentences to better explain how the cuttoff value of 15 was chosen for patient regret, although it is detailed in the refferenced articles, ensuring some additional information regarding this aspext would improve the quality of the methods section;

Reply: The materials and methods section has been amended to explain the selection of >15 as the cutoff defining critical patient’s regret (lines 167-170).  

2) the abbreviation for Clavien-Dindo Classification is not mentioned at the first use on line 151, although it is used in the following paragraphs as CDC;

Reply: Thanks to the reviewer for this important comment. The abbreviation CDC has been added to the manuscript as suggested (lines 158-207).

3) the explanation for ISUP is provided on line 191, please move this on line 169 where it is first used. 

Reply: The manuscript has been revised to ensure that the explanation for ISUP has now been added to the manuscript at its first appearance (lines 178 and 201).